# Neural Attention Distillation: Erasing Backdoor Triggers from Deep Neural Networks

**Yige Li[1]  Xixiang Lyu[1]†  Nodens Koren[2]  Lingjuan Lyu[3]  Bo Li[4]  Xingjun Ma[5]†**
[1]Xidian University  [2]The University of Melbourne  [3]Ant Group
[4]University of Illinois at Urbana–Champaign  [5]Deakin University, Geelong

## Abstract

Deep neural networks (DNNs) are known vulnerable to backdoor attacks, a training time attack that injects a trigger pattern into a small proportion of training data so as to control the model's prediction at the test time. Backdoor attacks are notably dangerous since they do not affect the model's performance on clean examples, yet can fool the model to make incorrect prediction whenever the trigger pattern appears during testing. In this paper, we propose a novel defense framework Neural Attention Distillation (NAD) to erase backdoor triggers from backdoored DNNs. NAD utilizes a teacher network to guide the finetuning of the backdoored student network on a small clean subset of data such that the intermediate-layer attention of the student network aligns with that of the teacher network. The teacher network can be obtained by an independent finetuning process on the same clean subset. We empirically show, against 6 state-of-the-art backdoor attacks, NAD can effectively erase the backdoor triggers using only 5% clean training data without causing obvious performance degradation on clean examples. Our code is available at https://github.com/bboylyg/NAD.

## 1  Introduction

In recent years, deep neural networks (DNNs) have been widely adopted into many important real-world and safety-related applications. Nonetheless, it has been demonstrated that DNNs are prone to potential threats in multiple phases of their life cycles. A type of well-studied adversary is called the adversarial attack (Szegedy et al., 2013; Goodfellow et al., 2014; Ma et al., 2018; Jiang et al., 2019; Wang et al., 2019b; 2020; Duan et al., 2020; Ma et al., 2020). At test time, state-of-the-art DNN models can be fooled into making incorrect predictions with small adversarial perturbations (Madry et al., 2018; Carlini & Wagner, 2017; Wu et al., 2020; Jiang et al., 2020). DNNs are also known to be vulnerable to another type of adversary known as the backdoor attack. Recently, backdoor attacks have gained more attention due to the fact it could be easily executed in real scenarios (Gu et al., 2019; Chen et al., 2017). Intuitively, backdoor attack aims to trick a model into learning a strong correlation between a trigger pattern and a target label by poisoning a small proportion of the training data. Even trigger patterns as simple as a single pixel (Tran et al., 2018) or a black-white checkerboard (Gu et al., 2019) can grant attackers full authority to control the model's behavior.

Backdoor attacks can be notoriously perilous for several reasons. First, backdoor data could infiltrate the model on numerous occasions including training models on data collected from unreliable sources or downloading pre-trained models from untrusted parties. Additionally, with the invention of more complex triggers such as natural reflections (Liu et al., 2020b) or invisible noises (Liao et al., 2020; Li et al., 2019; Chen et al., 2019c), it is much harder to catch backdoor examples at test time. On top of that, once the backdoor triggers have been embedded into the target model, it is hard to completely eradicate their malicious effects by standard finetuning or neural pruning (Yao et al., 2019; Li et al., 2020b; Liu et al., 2020b). A recent work also proposed the mode connectivity repair (MCR) to remove backdoor related neural paths from the network (Zhao et al., 2020a). On the other hand, even though detection-based approaches have been performing fairly well on identifying backdoored models (Chen et al., 2019a; Tran et al., 2018; Chen et al., 2019b; Kolouri et al., 2020), the identified backdoored models still need to be purified by backdoor erasing techniques.

---

†Correspondence to: Xixiang Lyu (xxlv@mail.xidian.edu.cn), Xingjun Ma (daniel.ma@deakin.edu.au)

In this work, we propose a novel backdoor erasing approach, *Neural Attention Distillation* (*NAD*), for the backdoor defense of DNNs. NAD is a distillation-guided finetuning process motivated by the ideas of knowledge distillation (Bucilua et al., 2006; Hinton et al., 2014) and neural attention transfer (Zagoruyko & Komodakis, 2017; Huang & Wang, 2017; Heo et al., 2019). Specifically, NAD utilizes a *teacher* network to guide the finetuning of a *backdoored student* network on a small subset of clean training data so that the intermediate-layer attention of the student network is well-aligned with that of the teacher network. The teacher network can be obtained from the backdoored student network via standard finetuning using the same clean subset of data. We empirically show that such an attention distillation step is far more effective in removing the network's attention on the trigger pattern in comparison to the standard finetuning or the neural pruning methods.

Our main contributions can be summarized as follows:

- We propose a simple yet powerful backdoor defense approach called Neural Attention Distillation (NAD). NAD is by far the most comprehensive and effective defense against a wide range of backdoor attacks.
- We suggest that attention maps can be used as an intuitive way to evaluate the performance of backdoor defense mechanisms due to their ability to highlight backdoored regions in a network's topology.

## 2   RELATED WORK

**Backdoor Attack.** Backdoor attack is a type of attack emerging in the training pipeline of DNNs. Oftentimes, a backdoor attack is accomplished by designing a trigger pattern with (poisoned-label attack) (Gu et al., 2019; Chen et al., 2017; Liu et al., 2018b) or without (clean-label attack) (Shafahi et al., 2018; Turner et al., 2019; Liu et al., 2020b) a target label injected into a subset of training data. These trigger patterns can appear in forms as simple as a single pixel (Tran et al., 2018) or a tiny patch (Chen et al., 2017), or in more complex forms such as sinusoidal strips (Barni et al., 2019) and dynamic patterns (Li et al., 2020c; Nguyen & Tran, 2020). Trigger patterns may also appear in the form of natural reflection (Liu et al., 2020b) or human imperceptible noise (Liao et al., 2020; Li et al., 2019; Chen et al., 2019c), making them more stealthy and hard to be detected even by human inspection. Recent studies have shown that a backdoor attack can be conducted even without access to the training data (Liu et al., 2018b) or in federated learning (Xie et al., 2019; Bagdasaryan et al., 2020; Lyu et al., 2020). Surveys on backdoor attacks can be found in (Li et al., 2020a; Lyu et al., 2020).

**Backdoor Defense.** Existing works primarily focused on two types of strategies to defend against backdoor attacks. Depending on the methodologies, a backdoor defense can be either backdoor detection or trigger erasing.

*Detection-based methods* aim at identifying the existence of backdoor adversaries in the underlying model (Wang et al., 2019a; Kolouri et al., 2020) or filtering the suspicious samples from input data for re-training (Tran et al., 2018; Gao et al., 2019; Chen et al., 2019b). Although these methods have been performing fairly well on distinguishing whether a model has been poisoned, the backdoor effects still remain in the backdoored model. On the other hand, *Erasing-based methods* aim to directly purify the backdoored model by removing the malicious impacts caused by the backdoor triggers, while simultaneously maintain the model's overall performance on clean data. A straightforward approach is to directly finetune the backdoored model on a small subset of clean data, which is typically available to the defender (Liu et al., 2018b). Nonetheless, training on only a small clean subset can lead to *catastrophic forgetting* (Kirkpatrick et al., 2017), where the model overfits to the subset and consequently causes substantial performance degradation. Fine-pruning (Liu et al., 2018a) alleviates this issue by pruning less informative neurons prior to finetuning the model. In such a way, the standard finetuning process can effectively erase the impact of backdoor triggers without significantly deteriorating the model's overall performance. WILD (Liu et al., 2020a) proposed to utilize data augmentation techniques alongside distribution alignment between clean samples and their occluded versions to remove backdoor triggers from DNNs. Other techniques such as regularization (Truong et al., 2020) and mode connectivity repair (Zhao et al., 2020a) have also been explored to mitigate backdoor attacks. While promising, existing backdoor erasing methods still suffer from a number of drawbacks. Efficient methods can be evaded by the latest attacks (Liu et al., 2018a; 2020b), whereas effective methods are typically computationally expensive (Zhao et al., 2020a). In this work, we propose a novel finetuning-based backdoor erasing approach that is not only effective but efficient against a wide range of backdoor attacks.

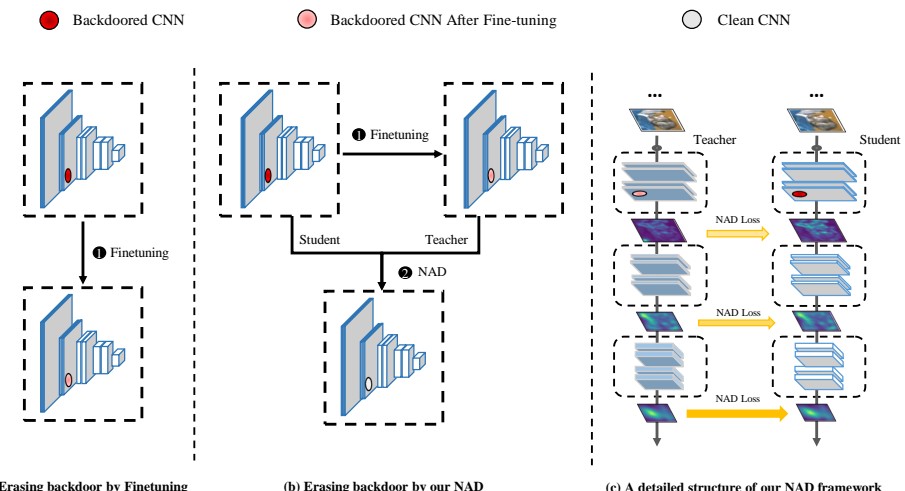

Figure 1: The pipeline of backdoor erasing techniques. (a) The standard finetuning process, (b) our proposed NAD approach, and (c) our NAD framework using ResNet (He et al., 2016) as an example. NAD erases backdoor trigger following a two-step procedure: 1) obtain a teacher network by finetuning the backdoored network with a subset of clean training data, then 2) combine the teacher and the student through the neural attention distillation process. The attention representations are computed after each residual group, and the NAD distillation loss is defined in terms of the attention representations of the teacher and the student networks.

**Knowledge Distillation (KD).** KD was first proposed to compress a bigger or an ensemble of well-trained network(s) into a compact smaller network (Bucilua et al., 2006; Hinton et al., 2014). In this process, the more knowledgeable network is referred to as the *teacher network*, and the smaller network is the *student network*. Feature maps and attention mechanisms have been demonstrated to be very useful in KD to supervise the training of student networks (Romero et al., 2015; Zagoruyko & Komodakis, 2017; Huang & Wang, 2017; Song et al., 2018; Ahn et al., 2019; Heo et al., 2019). They can help the student network to learn more high-quality intermediate representations, leading to an improved distillation effect and better student network performance (Romero et al., 2015; Zagoruyko & Komodakis, 2017). KD has also shown its potential in other fields such as adversarial robustness (Papernot et al., 2016), multi-granularity lip reading (Zhao et al., 2020b), and data augmentation (Bagherinezhad et al., 2018). In this work, we propose a new backdoor defense technique based on the combination of knowledge distillation and neural attention transfer.

## 3 PROPOSED APPROACH

In this section, we first describe the defense setting, then introduce the proposed NAD approach.

**Defense Setting.** We adopt a typical defense setting where the defender outsourced a backdoored model from an untrusted party and is assumed to have a small subset of clean training data to finetune the model. The goals of backdoor erasing are to erase the backdoor trigger from the model while retaining the performance of the model on clean samples.

### 3.1 NEURAL ATTENTION DISTILLATION

**Overview.** We illustrate the differences between NAD and the traditional finetuning approach in Figure 1. Instead of using the finetuned network directly as our final model, we employ it as a teacher network and use it in conjunction with the original backdoored network (i.e. student network) through an attention distillation process. The job of NAD is to align neurons that are more responsive to the trigger pattern with benign neurons that only responsible for meaningful representations. The main challenge for NAD is thus to find the proper attention representations to distill. Over the next few subsections, we will define the attention representation used throughout our work formally and introduce the loss functions used in the process of attention distillation.

**Attention Representation.** Given a DNN model $F$, we denote the activation output at the $l$-th layer as $F^l \in \mathbb{R}^{C \times H \times W}$ with $C$, $H$ and $W$ being the dimensions of the channel, the height, and the width

of the activation map respectively. We define $\mathcal{A} : \mathbb{R}^{C \times H \times W} \to \mathbb{R}^{H \times W}$ to be an attention operator that maps an activation map to an attention representation. Specifically, $\mathcal{A}$ takes a 3D activation map $F$ as input and outputs a flattened 2D tensor along the channel dimension. We explore three possible formulations of the attention operator $\mathcal{A}$ as suggested in (Zagoruyko & Komodakis, 2017):

$$\mathcal{A}_{\text{sum}}(F^l) = \sum_{i=1}^{C} \left| F_i^l \right| ; \quad \mathcal{A}_{\text{sum}}^p(F^l) = \sum_{i=1}^{C} \left| F_i^l \right|^p ; \quad \mathcal{A}_{\text{mean}}^p(F^l) = \frac{1}{C} \sum_{i=1}^{C} \left| F_i^l \right|^p , \qquad (1)$$

where $F_i^l$ is the activation map of the $i$-th channel, $|\cdot|$ is the absolute value function and $p > 1$. Intuitively, $\mathcal{A}_{\text{sum}}$ reflects all activation regions including both the benign and the backdoored neurons. $\mathcal{A}_{sum}^p$ is a generalized version of $\mathcal{A}_{\text{sum}}$ that amplifies the disparities between the backdoored neurons and the benign neurons by an order of $p$. In other words, the larger the $p$ is, the more weight is placed on the parts with highest neuron activations. $\mathcal{A}_{\text{mean}}$ aligns the activation center of the backdoored neurons with that of the benign neurons by taking the mean over all activation regions. An empirical understanding of the three attention representations is provided in the experiments.

**Attention Distillation Loss.** A detailed structure of our NAD framework is illustrated in Figure 1(c). For ResNets (He et al., 2016), we compute attention representations using one of the proposed attention functions after each group of residual blocks. The teacher network is kept fixed throughout the distillation process. The distillation loss at the $l$-th layer of the network is defined in terms of the teacher's and the student's attention maps:

$$\mathcal{L}_{\text{NAD}}\left(F_T^l, F_S^l\right) = \left\| \frac{\mathcal{A}(F_T^l)}{\left\| \mathcal{A}(F_T^l) \right\|_2} - \frac{\mathcal{A}(F_S^l)}{\left\| \mathcal{A}(F_S^l) \right\|_2} \right\|_2 , \qquad (2)$$

where $\|\cdot\|_2$ is the $L_2$ norm and $\mathcal{A}(F_T^l)/\mathcal{A}(F_S^l)$ is the computed attention maps of the teacher/student network. It is worth mentioning that the normalization of the attention map is crucial to a successful distillation (Zagoruyko & Komodakis, 2017).

**Overall Training Loss.** The overall training loss is a combination of the cross entropy (CE) loss and the sum of the Neural Attention Distillation (NAD) loss over all $K$ residual groups:

$$\mathcal{L}_{total} = \mathbb{E}_{(\boldsymbol{x}, y) \sim \mathcal{D}}[\mathcal{L}_{\text{CE}}(F_S(\boldsymbol{x}), y) + \beta \cdot \sum_{l=1}^{K} \mathcal{L}_{\text{NAD}}(F_T^l(\boldsymbol{x}), F_S^l(\boldsymbol{x}))], \qquad (3)$$

where $\mathcal{L}_{\text{CE}}(\cdot)$ measures the classification error of the student network, $\mathcal{D}$ is a subset of clean data used in finetuning, $l$ is the index of the residual group, and $\beta$ is a hyperparameter controlling the strength of the attention distillation.

Before the attention distillation, a teacher network should be in place. We finetune the backdoored student network on the same clean subset $\mathcal{D}$ to obtain the teacher network. We will investigate how the choice of the teacher network affects the performance of NAD in Section 4.4. We only distill the student network once as our NAD approach is effective enough to remove the backdoor by only a few epochs of distillation. A comprehensive analysis of iteratively applied NAD is also provided in the Appendix G.

## 4 EXPERIMENTS

In this section, we first introduce the experimental setting. We then evaluate and compare the effectiveness of NAD with 3 existing backdoor erasing methods on 6 state-of-the-art backdoor attacks. Finally, we provide a comprehensive understanding of NAD.

### 4.1 EXPERIMENTAL SETTING

**Backdoor Attacks and Configurations.** We consider 6 state-of-the-art backdoor attacks: 1) Bad-Nets (Gu et al., 2019), 2) Trojan attack (Liu et al., 2018b), 3) Blend attack (Chen et al., 2017), 4) Clean-label attack(CL) (Turner et al., 2019) , 5) Sinusoidal signal attack(SIG) (Barni et al., 2019), and 6) Reflection attack(Refool) (Liu et al., 2020b). For a fair evaluation, we follow the configuration, including the trigger patterns, the trigger sizes and the target labels, of these attacks in their original papers. We test the performance of all attacks and erasing methods on two benchmark datasets, CIFAR-10 and GTSRB, with WideResNet (WRN-16-1*) being the base model throughout the experiments. More details on attack configurations are summarized in Appendix A.

---

*https://github.com/szagoruyko/wide-residual-networks

Table 1: Performance of 4 backdoor defense methods against 6 backdoor attacks evaluated using the attack success rate (ASR) and the classification accuracy (ACC). The *deviation* indicates the % changes in ASR/ACC compared to the baseline (i.e. no defense). The experiments for Refool were done on GTSRB, while all other experiments were done on CIFAR-10. The best results are in **bold**.

| Backdoor Attack | Before | | Finetuning | | Fine-pruning | | MCR (t = 0.3) | | NAD (Ours) | |
|---|---|---|---|---|---|---|---|---|---|---|
| | ASR | ACC | ASR | ACC | ASR | ACC | ASR | ACC | ASR | ACC |
| BadNets | 100 | 85.65 | 17.18 | **81.22** | 99.73 | 81.14 | **4.65** | 80.94 | 4.77 | 81.17 |
| Trojan | 100 | 81.24 | 71.76 | 77.88 | 41.00 | 78.17 | 41.25 | 78.76 | **19.63** | **79.16** |
| Blend | 99.97 | 84.95 | 36.60 | 81.22 | 93.62 | 81.13 | 64.33 | 80.34 | **4.04** | **81.68** |
| CL | 99.21 | 82.43 | 75.08 | **81.73** | 29.88 | 79.32 | 32.95 | 79.04 | **9.18** | 80.34 |
| SIG | 99.91 | 84.36 | 9.18 | 81.28 | 74.26 | 81.60 | **1.62** | 80.94 | 2.52 | **81.95** |
| Refool | 95.16 | 82.38 | 14.38 | 80.34 | 63.49 | 80.64 | 8.76 | 78.84 | **3.18** | **80.73** |
| Average | 99.04 | 83.50 | 37.36 | 80.61 | 67.00 | 80.50 | 25.59 | 79.81 | **7.22** | **80.83** |
| Deviation | - | - | ↓61.68 | ↓2.89 | ↓32.04 | ↓3 | ↓73.44 | ↓3.69 | ↓**91.82** | ↓**2.66** |

**Defense Configuration.** We compare our NAD approach with 3 existing backdoor erasing methods: 1) the standard finetuning, 2) Fine-pruning (Liu et al., 2018a), and 3) mode connectivity repair (MCR) (Zhao et al., 2020a). We assume all defense methods have access to the same 5% of the clean training data.

For NAD, we finetune the backdoored model (i.e. the student network) on the 5% accessible clean data for 10 epochs (results for 20 epochs can be found in Appendix J) using the Stochastic Gradient Descent (SGD) optimizer with a momentum of 0.9, an initial learning rate of 0.1, and a weight decay factor of $10^{-4}$. The learning rate is divided by 10 after every 2 epochs. We use a batch size of 64, and apply typical data augmentation techniques including random crop (padding = 4), horizontal flipping, and Cutout (n_holes=1 and length=9) (DeVries & Taylor, 2017). The data augmentations are applied to each batch of training images at each training iteration, following a typical DNN training process. The same data augmentations are also applied to other fintuning-based baseline methods. Additionally, we have a comparison of our NAD method to just using data augmentations Cutout and Mixup in Appendix B. For the distillation loss, we compute the attention maps using the $\mathcal{A}_{sum}^2$ attention operator after the three groups of residual blocks of WRN-16-1 (see Figure 1(c)) . An extensive study on four attention functions is given in Section 4.3. For the hyperparameter $\beta$, we adaptively set it to different values for each backdoor attack. We provide more details on defense settings in Table 3 (see Appendix A).

**Evaluation Metrics.** We evaluate the performance of defense mechanisms with two metrics: attack success rate (ASR), which is the ratio of backdoored examples that are misclassified as the target label, and model's accuracy on clean samples (ACC). The more the ASR drops and the less the ACC drops, the stronger the defense mechanism is.

## 4.2 Effectiveness of Our NAD Defense

In order to assess the effectiveness of our proposed NAD defense, we evaluate its performance against 6 backdoor attacks using two metrics (i.e. ASR and ACC). We then compare the performance of NAD with the other 3 existing backdoor defense methods in Table 1. Our experiment shows that our NAD defense remarkably brought the average ASR from nearly 100% down to 7.22%. In comparison, Finetuning, Fine-pruning, and MCR are only able to reduce the average ASR to 37.36%, 67.00%, and 25.59% respectively.

MCR has an erasing effect stronger than that of the NAD's by 0.12% on BadNets and by 0.9% on SIG, but its performance against the 4 other attacks are much poorer. Specifically, MCR failed to defend 60.29% more attack on Blend, 23.77% more attack on CL, and 18.77% more attack on Trojan in comparison to NAD. Our hypothesis on this is that backdoor triggers with much complicated adversarial noises (e.g. random or PGD perturbations) would hinder the training of connect path and consequently gives MCR a hard time at finding robust models against these backdoor attacks. Interestingly, finetuning did moderately well in erasing all kinds of attacks. A reasonable explanation to this is that the data augmentation techniques used in the early stage might have recovered some trigger-related images leading the backdoor model to unlearn the original trigger. For instance, the black edges in the zero-padding of images may have effects similar to the black-white trigger in BadNets. On the other hand, Fine-pruning gives a poor performance on mitigating all backdoor

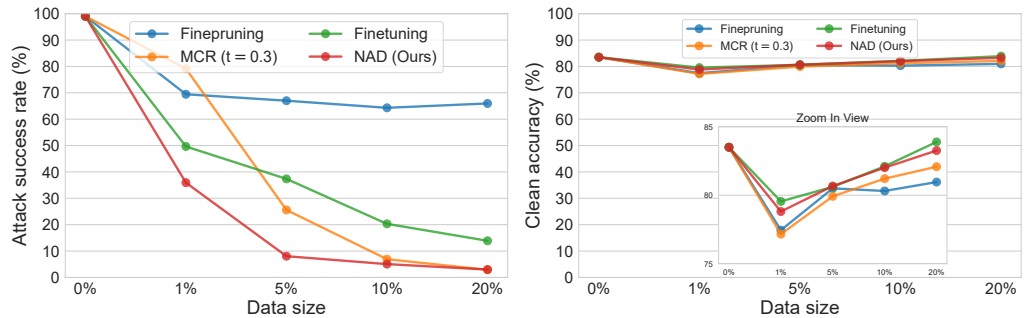

Figure 2: Performance of 4 backdoor erasing methods under different % of available clean data. The plots show the average ASR (left) and ACC (right) over all 6 attacks. NAD significantly reduces the ASR to nearly 0% with 20% clean data.

attacks under our experiment settings. We speculate that a potential reason behind this is due to the low number of neurons in the last convolutional layer of WRN-16-1, indicating a high mixing of benign neurons and backdoored neurons. This may cause a significant reduction in the classification accuracy and in consequence makes the pruning ineffective.

In summary, all erasing methods have some negative effects on the ACC, but the drops by utilizing NAD is the least prominent (merely 2.66%). More comparisons to data augmentation techniques and the effectiveness in erasing all-target and adaptive backdoor attacks can be found in Appendix B, H and Appendix K respectively.

**Effectiveness under Different Percentages of Clean Data.** We are also interested in studying the correlation between the performance of NAD and the amount of available pristine data. Intuitively, we anticipate NAD to be stronger when we have more clean training data, and vice versa. The performance of NAD and 3 other defense mechanisms with various sizes of "purifying dataset" is recorded in Figure 2.

It is within our expectation that both MCR and our proposed NAD approach are capable of defending against all 6 backdoor attacks almost 100% of the time when 20% of clean training data are available to us. Nonetheless, NAD still beats MCR in terms of the convergence rate. We will show in Appendix C that our proposed NAD approach converges much faster than MCR. Additionally, we find that finetuning becomes much more effective when the backdoor model is retrained on 20% of the clean data. Despite the improvement, the standard finetuning method is still considerably worse than MCR and NAD by ∼10% in terms of the ASR. Surprisingly, Fine-pruning gains almost no benefits from clean data additional to the 5% used in the first experiment. This is because the benign neurons and the backdoored neurons are highly blended in the last layer of the backdoored network, leading to excessive pruning of the backdoored neurons. Retraining becomes pointless when too many backdoored neurons have been removed.

In short, even with just 1% of clean training data available, our NAD can still effectively bring the average ASR from 99.04% down to 35.93%, while only sacrifices 4.69% of ACC.

**Comparison to Trigger Recovering.** Some existing works proposed defense methods that predict the distributions of backdoor triggers through generative modeling or neuron reverse engineering. The predicted distributions can subsequently be sampled to craft *backdoored data with correct labels*. These "remedy data" can subsequently be used in retraining to alleviate the impact of backdoor triggers (Wang et al., 2019a; Qiao et al., 2019).

In this subsection, we compare the performance of our NAD defense to one such approach, MESA (Qiao et al., 2019), which is the current state-of-the-art trigger recovering method. Specifically, we are interested in comparing our method to the retraining-based method in general. To do this, we first retrain the backdoored model separately using the trigger generated by MESA (Rec-T) and the original trigger (Org-T). We then evaluate the performance of these retrained models and compare their performances to that of the NAD's. The results are presented in Table 6 (see Appendix D).

The retraining-based approach is strong at defending against the BadNets attack. It is able to reduce the ASR from 100% to under 5% without significantly sacrificing the ACC. Nonetheless, its performance is nowhere close to our NAD defense when facing the CL attack. This is a good indication that the backdoored neurons can be fixed by retraining with the remedy data when the backdoor behavior is induced by only a few backdoored neurons. On the other hand, it is too much for these

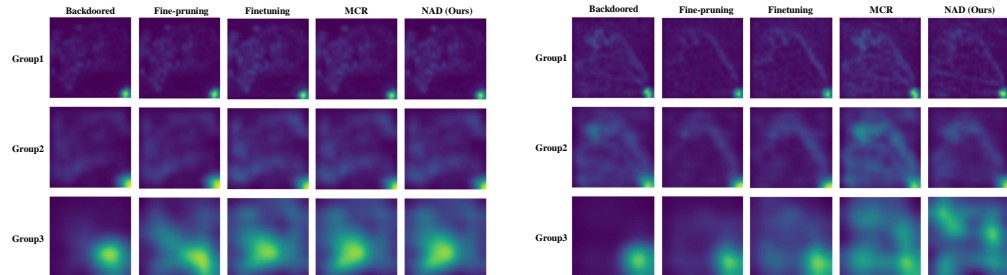

Figure 3: Visualization of the attention maps learned at each residual group of the WRN-16-1 by different defense methods for a BadNets (left) or CL (right) backdoored image (see Appendix A). Our NAD method demonstrates a more effective erasing effect at the deeper layers (e.g. Group 3).

remedy data to handle backdoors obtained through the combination of complex adversarial noises and stronger trigger patterns.

### 4.3 A Comprehensive Understanding and Analysis of NAD

In this section, we first provide intuition behind what an attention map is from the visual perspective, we then compare the efficacy of various choices of attention representation as promised in Section 3.1. Finally, we explore the adjustment border of the hyperparameter $\beta$.

**Understanding Attention Maps.** The activation information of all neurons in a layer of a neural network can be referred from the attention map of that layer. The conjunct of all attention maps hence reflects the most discriminative regions in the network's topology (Lopez et al., 2019). To give intuition on how attention maps help NAD with erasing backdoor triggers, we visualize and compare the attention maps before and after backdoor erasing in Figure 3. We use the attention function $\mathcal{A}^2_{sum}$ to derive the attention maps. For the backdoored models, all attention maps are completely (i.e. group1, group2, and group3) biased towards the backdoor trigger region, implying that the backdoor trigger can easily mislead the network to misbehave. The objective of backdoor erasing methods is consequently to relax the tension in this region. We show the results for two attacks, BadNets (left) and CL (right), to validate our hypothesis.

Attention maps can also be used as an indicator to deduce the performance of backdoor erasing methods. In Figure 3, we show the attention maps of a backdoored WRN-16-1 after purified by four different mechanisms: Fine-pruning, the standard finetuning, MCR, and our NAD approach. As expected, BadNets can easily be erased by the most defenses. To see why, refer to the attention maps of group 3; the network purified by the standard finetuning, MCR, and NAD pay almost no attention to the bottom right corner where the trigger is injected. For CL, only MCR and NAD are able to distract the backdoored model's from focusing on the triggered regions, which is also foreseeable as CL is a stronger attack. In addition, the activation intensity of NAD in the benign area (i.e. non-trigger area) is correspondingly greater than that of MCR, which also justifies our conclusion that NAD is better than MCR in erasing CL attacks from another perspective.

Next, we compare the performance of NAD under scenarios where 4 different attention functions, $\mathcal{A}_{mean}$, $\mathcal{A}^2_{mean}$, $\mathcal{A}_{sum}$, and $\mathcal{A}^2_{sum}$ are in place. We use the BadNets attack as our benchmark attack. Again, we evaluate the performance of NAD using two metrics, the ASR and the ACC, and the results are summarized in Appendix F. Despite all choices of attention function are able to help NAD erase backdoors quite efficiently, $\mathcal{A}^2_{sum}$ achieved the best overall results. A comparison to using the raw activation map for distillation can be found in Appendix I.

**Attention Distillation VS. Feature Distillation.** We outline two advantages of attention distillation over feature distillation: 1) *Integration*. The attention operators calculate the sum (or the mean) of the activation map over different channels (see Equation 1). It can thus provide an integrated measure of the overall trigger effect. On the contrary, the trigger effect may be scattered into different channels if we use the raw activation values directly. Such a difference between the attention and the feature maps is visualized in Figure 11 and 12 in Appendix I. Therefore, aligning the attention maps between the teacher and the student networks is more effective in weakening the overall trigger effect than directly aligning the raw feature maps. 2) *Regularization*. Due to its integration effect, an attention map contains the activation information of both backdoor-fired neurons and the benign neurons. This is important as the backdoor neurons can receive extra gradient information from the attention map even when they are not activated by the clean data. Moreover, attention maps have

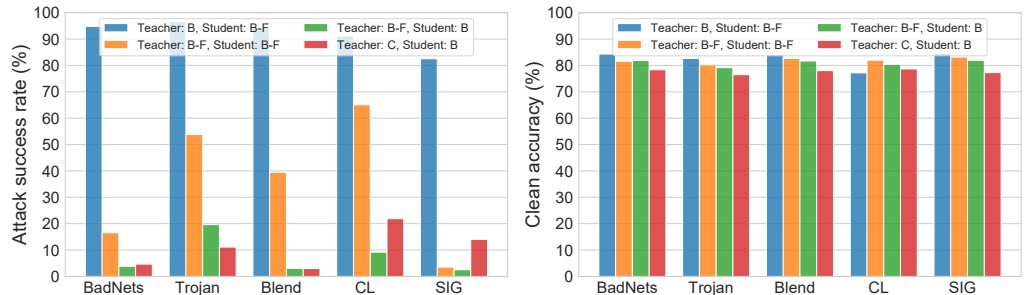

Figure 4: Comparison of 4 distillation combinations on CIFAR-10. The **B**, **B-F**, and **C** represent backdoored model, finetuned backdoored model, and model trained on the clean subset, respectively.

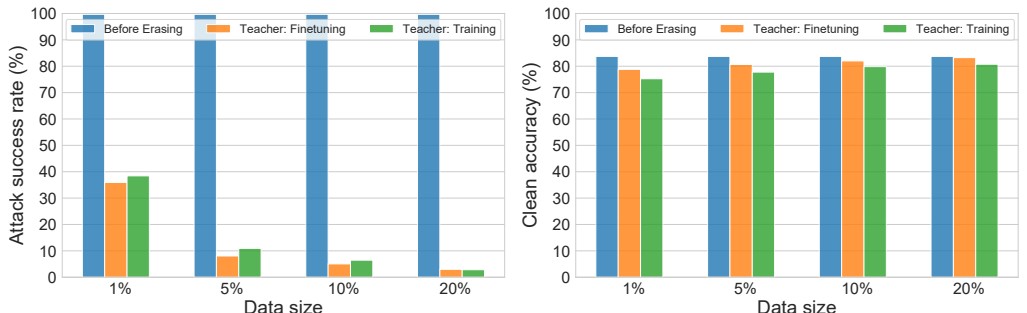

Figure 5: Performance of NAD with teachers trained on various % of clean CIFAR-10 data.

lower dimensions than feature maps. This makes the attention map based regularization (alignment) more easily to be optimized than feature map based regularization.

**Effect of Parameter** $\beta$**.** The selection of the distillation parameter $\beta$ is also a key factor for NAD to erase backdoor triggers successfully. We show the results of the coarse tuning $\beta$ for all the backdoor attacks in Appendix E, and it reveals that $\beta$ can certainly be tuned more to improve the performance of NAD. In short, the process of finding the right scaling factor $\beta$ is to find a balance between the ASR and the ACC. Even though bigger $\beta$ is more effective against backdoor attacks, Figure 8(right) in Appendix E shows that arbitrarily increasing $\beta$ may cause a degradation in the ACC. For instance, the purified network lost more than 50% of ACC when $\beta$ is set to 50,000. A practical strategy to select $\beta$ is to increase $\beta$ until the clean accuracy (right subfigure in Figure 8) drops below an acceptable threshold. This can reliably find an optimal $\beta$, as increasing $\beta$ can always improve the robustness (left subfigure in Figure 8).

### 4.4 FURTHER EXPLORATION OF NAD

Here, we explore how the combinations of teachers and students and the choice of the teacher affect the defense performance of NAD. For simplicity, we define **B** to be the backdoored network, **B-F** to be the finetuned backdoored network, and **C** to be the model trained from scratch using 5% of clean training data. Here we consider 4 combinations of networks: 1) **B** teacher and **B-F** student, 2) **B-F** teacher and **B** student, 3) **B-F** teacher and **B-F** student, and 4) **C** teacher and **B** student.

**Effect of Teacher-Student Combinations.** The ASR and ACC of NAD-purified networks using different teacher-students combinations are presented in Figure 4. We find that the ASR slightly diminished with the combination of **B** teacher and **B-F** student. On the contrary, when using a **B-F** teacher and a **B-F** student in the NAD framework, the ASR is drastically lowered (significantly improved defense). Interestingly, compared with the standard setting used throughout the previous experiments (i.e. Teacher: **B-F**, Student: **B**), using a trained-from-scratch model as the teacher can also reduce the average ASR by more than 80%. Furthermore, this combination works even better in tackling the Trojan attack. Nonetheless, the combination of **C** teacher and **B** student significantly deteriorates the ACC of the purified model, which is also predictable because the amount of clean data available to train a model from scratch has a direct effect on the model's accuracy. It is hence not a surprise that an incompetent teacher misleads the student network into learning flawed features.

Table 2: Effectiveness of our NAD with different teacher architectures against BadNets on CIFAR-10. ASR: attack success rate; ACC: clean accuracy. The first column highlights the architectural difference between the teacher and the student network. The best results are **boldfaced**.

| Difference | Teacher | Student | Before | | NAD (Ours) | | Teacher |
|---|---|---|---|---|---|---|---|
| | | | ASR | ACC | ASR | ACC | ACC |
| Depth&Channel | WRN-10-2 | WRN-16-1 | 100% | 85.65% | 4.68% | 78.36% | 63.78% |
| Same | WRN-16-1 | WRN-16-1 | 100% | 85.65% | 4.55% | 74.53% | 61.21% |
| Channel | WRN-16-2 | WRN-16-1 | 100% | 85.65% | 3.04% | 78.68% | 64.25% |
| Depth | WRN-40-1 | WRN-16-1 | 100% | 85.65% | **2.95%** | 78.87% | 63.35% |
| Depth&Channel | WRN-40-2 | WRN-16-1 | 100% | 85.65% | 3.74% | **79.07%** | **64.53%** |

**Effect of the Choice of a Teacher.** Due to the competitive performance offered by the **C** teacher and **B** student combination in the last section, we are interested in exploring one question: *is the standard setting (i.e. **B-F** teacher and **B** student) we used throughout previous experiments still the best option when more clean data are available to us*? To answer this question, we compare the performance of both **B-F** teacher and **C** teacher to erase **B** student under various percentages of available clean data on CIFAR-10. We use the same training configurations described in Section 4.1. The results are shown in Figure 5. Compared with the finetuning option of **B-F** teacher, the **C** teacher trained on a subset of clean data also offers competitive performance. Specifically, in the case where we have 20% of clean data available to us, the training option reduces the ASR by 0.15% in comparison to the finetuning option; nonetheless, it reduces the ACC by a greater amount of 2.5%. This makes finetuning still a better option even when we have 20% clean data at hands.

**Effectiveness of Different Teacher Architectures.** This experiment is conducted on CIFAR-10 against BadNets attack. We consider 5 teacher architectures: WRN-10-2, WRN-16-1, WRN-16-2, WRN-40-1 and WRN-40-2. We fix the student network to WRN-16-1. Since the teachers are of different (except one) architectures as the student, we train the teacher networks from scratch using only 5% clean training data. Our NAD is applied following the same configurations in Section 4.1. The results are reported in Table 2. We can see that all 5 teacher networks are able to purify the backdoored student effectively under our NAD framework. NAD can reduce the ASR from 100% to 4.68% even when a small teacher WRN-10-2 is used. This confirms that our NAD defense generalizes well across different network architectures. Note that the clean accuracy of NAD decreases more than in the same architecture setting (see Table 1). This is because the teachers have lower clean accuracy when trained from scratch on only 5% clean data, as we have analyzed in Figure 4.

**Why a finetuned teacher can purify a backdoored student?** During the NAD process, the backdoored neurons in the finetuned teacher network are less likely to be activated by the clean finetuning data. Moreover, as shown in Figure 12 (Appendix I), finetuning can suppress the trigger effect, and at the same time, boost the benign neurons (more visible light green regions in Figure 12 (b)). By using the squared sum attention map $\mathcal{A}_{sum}^2$ (analyzed in Section 4.3 and Appendix I), this trigger erasing effect can accumulate from the already finetuned teacher network, leading to more robust and cleaner student network. Note that the student can still overfit to the partially purified teacher if it is overly finetuned.

## 5 CONCLUSION

In this work, we proposed a novel knowledge distillation based backdoor defense framework Neural Attention Distillation (NAD). We demonstrated empirically that our proposed approach is able to achieve a superior performance against 6 state-of-the-art backdoor attacks in comparison to 3 other backdoor defense methods. Additionally, we propose the use of attention maps as an intuitive way to evaluate the performance of backdoor defense mechanisms due to their capability of displaying backdoored regions in a network's topology visually. On top of that, we explored how different experimental settings might affect our proposed method. Empirical results demonstrate that our results are fairly resilient to the changes in experimental settings and thus can be conveniently employed without exhaustive hyperparameter tuning. Overall, our proposed NAD backdoor defense framework provides a strong baseline in mitigating the backdoor threat in model deployment.

## ACKNOWLEDGEMENT

This work is supported by China National Science Foundation under grant number 62072356 and Amazon research award.

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

# A    MORE IMPLEMENTATION DETAILS

The backdoor triggers used in our experiments are shown in Figure 6.

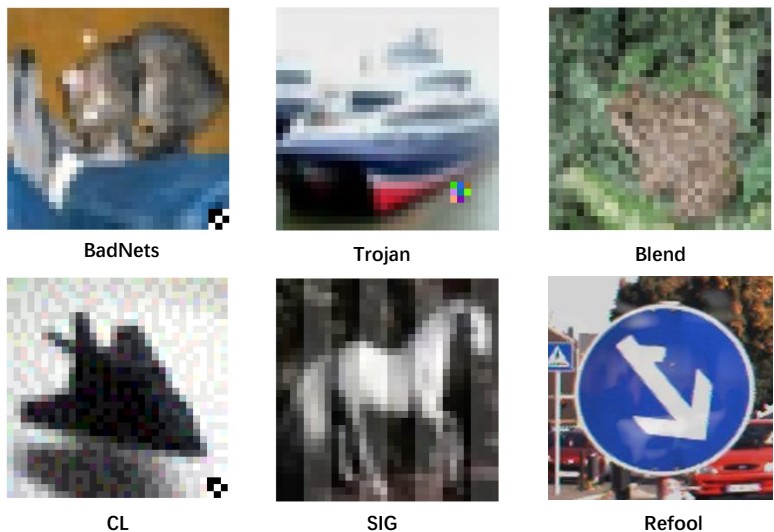

Figure 6: Examples of backdoored CIFAR-10 images by the 6 attacks.

Table 3: A configuration summary for the 6 backdoor attacks: datasets, models, and triggers.

| Backdoork | BadNets | Trojan | Blend | Clean-Label | Signal | Refool |
|---|---|---|---|---|---|---|
| Dataset | CIFAR-10 | CIFAR-10 | CIFAR-10 | CIFAR-10 | CIFAR-10 | GTSRB |
| Model | WideResNet | WideResNet | WideResNet | WideResNet | WideResNet | WideResNet |
| Inject Rate | 0.1 | 0.05 | 0.1 | 0.08 | 0.08 | 0.08 |
| Trigger Type | Grid | Square | Random Noise | Grid + PGD Noise | Sinusoidal Signal | Reflection |
| Trigger Size | $3 \times 3$ | $3 \times 3$ | Full Image | $3 \times 3$ | Full Image | Full Image |
| Target Label | 0 | 0 | 0 | 0 | 0 | 0 |
| ASR | 100.00% | 100.00% | 99.97% | 99.21% | 99.91% | 95.16% |
| ACC | 85.65% | 81.24% | 84.95% | 82.43% | 84.36% | 82.38% |

Detailed implementation on 6 state-of-the-art backdoor attacks:

- BadNets: The trigger is a $3 \times 3$ checkerboard (pixel values are 128 or 255) at the bottom right corner of images. We labeled the backdoor examples with a chosen target label and achieved an attack success rate of 100% with an injection rate of 10%.

- Trojan attack. We follow the method proposed in the paper to reverse engineer a $3 \times 3$ square trigger from the last fully-connected layer of the network. In order to reduce the impact on clean accuracy, we poisoned only 5% of training data with the reverse-engineered Trojan trigger. We achieved an attack success rate of 100% with an injection rate of 5%.

- Blend attack. We used the random patterns reported in the original paper. We achieved an attack success rate of 99.97% with an injection rate of 10% and a blend ratio of $\alpha = 0.2$.

- Clean-label attack (CL). We followed the same settings as reported in the paper. Specifically, we used Projected Gradient Descent (PGD) to generate adversarial perturbations bounded to $L_\infty$ maximum perturbation $\epsilon = 0.15$. The trigger is a $3 \times 3$ grid at the bottom right corner of images. We achieved an attack success rate of 99.21% with an injection rate of 8%.

- Sinusoidal signal attack (SIG). We generate the backdoor trigger following the horizontal sinusoidal function defined in their paper with $\Delta = 20$ and $f = 6$. We achieved an attack success rate of 99.91% with an injected rate of 8%.

- Reflection attack (Refool). The implementation is based on the open-source code[†]. We achieved an attack success rate of 95.16% with an injection rate of 8%.

**More Details on Defense Baselines** We adopted the same settings used in NAD for the standard finetuning approach and finetuned the model until convergence. We replicated Fine-pruning[‡] via PyTorch and pruned the last convolutional layer of the model as suggested in the original paper Liu et al. (2018a). For a fair comparison, the pruning ratio was set to a value such that the ACC of the pruned network matched the ACC of our NAD approach. We used the open-source code[§] for mode connectivity repair (MCR) and set the endpoint model t = 0 and t = 1 with the same backdoored WRN-16-1. We trained the connection path for 100 epochs and evaluated the defense performance of the model on the path. Other settings of the code remain unchanged.

## B    COMPARISON WITH DATA AUGMENTATION TECHNIQUES

Cutout (DeVries & Taylor, 2017) and Mixup (Zhang et al., 2018) are popular data augmentation methods for CNNs. Cutout masks out random sections of input images during training and Mixup randomly morphs the training images. We evaluate in this section the independent effectiveness of Mixup and Cutout in erasing backdoor triggers. For Cutout[¶], we set the number of patches to be cut out of each image to 1 and each patch is a $3 \times 3$ square. For Mixup[‖], we set $\alpha$ to be the default value of 1, indicating that we sample the weight uniformly between zero and one. Other settings for attacks and defenses are identical to the settings specified in Section 4.1. The results (see Table 4) can be a supplement of Table 1. We conclude that data augmentation techniques have mitigating effects on backdoors only when the transformation images are similar to the trigger patterns. They are hence not general against a wide range of backdoor attacks.

Table 4: Comparison with Mixup and Cutout on erasing backdoor triggers.

| Backdoor Attack | Before | | Mixup | | Cutout | | NAD (Ours) | |
|---|---|---|---|---|---|---|---|---|
| | ASR | ACC | ASR | ACC | ASR | ACC | ASR | ACC |
| BadNets | 100% | 85.65% | 68.22% | 80.27% | 28.17% | **82.73**% | **3.81**% | 81.85% |
| Trojan | 100% | 81.24% | 96.20% | 71.83% | 50.22% | **80.13**% | **19.63**% | 79.16% |
| Blend | 99.97% | 84.95% | 99.11% | 80.51% | 15.30% | **81.78**% | **3.04**% | 81.68% |
| CL | 99.21% | 82.43% | 93.77% | 77.13% | 73.33% | **81.34**% | **9.18**% | 80.34% |
| SIG | 99.91% | 84.36% | 52.11% | 79.94% | 99.95% | **82.77**% | **2.52**% | 81.95% |
| Refool | 95.16% | 82.38% | 8.76% | 77.84% | 91.86% | 80.06% | **3.18**% | **80.73**% |
| Average | 99.04% | 83.50% | 69.69% | 77.92% | 59.80% | **81.46**% | **7.22**% | 80.83% |
| Deviation | - | - | ↓ 29.35% | ↓ 5.58% | ↓ 39.24% | ↓ **2.03**% | ↓ **91.82**% | ↓ 2.66% |

## C    MORE RESULTS OF MODE CONNECTIVITY REPAIR (MCR)

We use the open-source code of MCR and compare its performance to our NAD method. The experiments are conducted on CIFAR-10 dataset using 5% clean finetune data. We first run MCR with the two endpoint models t = 0 and t = 1 which use the same backdoored WRN-16-1 model. Figure 7 shows the convergence rate of MCR and our NAD against BadNets attack. We then run an additional experiment for MCR using two different endpoint models: t = 0 and t = 1 use the backdoored WRN-16-1 and the finetuned backdoored WRN-16-1 respectively. This result is reported in Table 5. We find that using different endpoint models can not further improve the performance of MCR.

---

[†]https://github.com/DreamtaleCore/Refool
[‡]https://github.com/kangliucn/Fine-pruning-defense
[§]https://github.com/IBM/model-sanitization/tree/master/backdoor/backdoor-cifar
[¶]https://github.com/uoguelph-mlrg/Cutout
[‖]https://github.com/leehomyc/mixup-pytorch

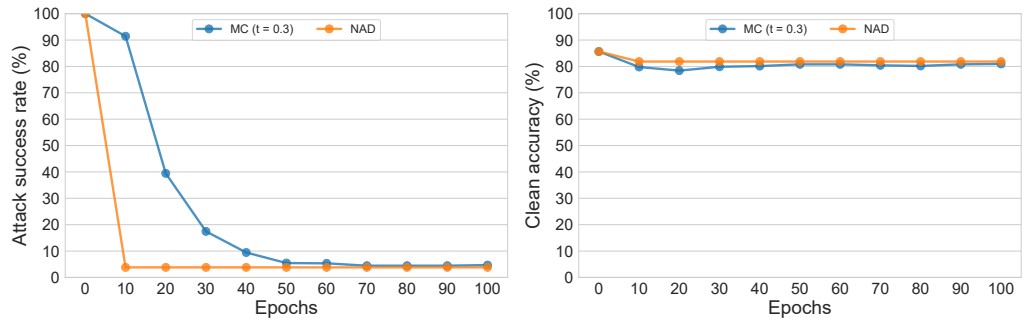

Figure 7: Convergence rate comparison between MCR and NAD against BadNets attack with 5% clean training data. We show the best result of MC at the connection point t = 0.3. Note that it takes longer for MCR to converge yet its ASR is still higher than that of the NAD's.

Table 5: Performance of MCR with different endpoint models on CIFAR-10 dataset. **B** denotes the backdoored WRN-16-1 and **F-B** denotes the backdoored WRN-16-1 after Fine-tuning. MCR-(B,B) denotes the default setting where the two endpoint models are both B, while MCR-(B,F-B) denotes the MCR using two different endpoint models B and F-B. The best results are **boldfaced**.

| Backdoor | Before | | MCR-(B,B) | | MCR-(B,F-B) | | NAD (Ours) | |
|---|---|---|---|---|---|---|---|---|
| Attack | ASR | ACC | ASR | ACC | ASR | ACC | ASR | ACC |
| BadNets | 100% | 85.65% | **4.65**% | 80.94% | 6.00% | 80.56% | 4.77% | **81.17**% |
| Trojan | 100% | 81.24% | 41.25% | 78.76% | 53.31% | 78.31% | **19.63**% | **79.16**% |
| Blend | 99.97% | 84.95% | 64.33% | 80.34% | 70.65% | 80.51% | **4.04**% | **81.68**% |
| CL | 99.21% | 82.43% | 32.95% | 79.04% | 42.66% | 80.31% | **9.18**% | **80.34**% |
| SIG | 99.91% | 84.36% | **1.62**% | 80.94% | 7.32% | 81.12% | 2.52% | **81.95**% |
| Refool | 95.15% | 82.38% | 8.76% | 78.84% | 10.95% | 79.03% | **3.18**% | **80.73**% |

# D EXPERIMENTAL RESULTS OF TRIGGER RECOVERING TECHNIQUE

Qiao et al. (2019) proposed MESA that recovers the trigger distribution via generative modeling and then removes the backdoor by model retraining. We implemented this work based on their open-source code[**]. Note that we report the best averaging results of defense performance and we changed nothing in the code besides setting the proportion of available training data to 5%. We present the results in Table 5.

Table 6: Comparison between NAD and retraining-based approaches that use both the original trigger (Org-T) and the MESA-recovered trigger (Rec-T). While all methods are able to reduce the ASR of BadNets to a similar level, NAD is able to reduce the ASR of CL by 16 more percent in comparison to the model retrained with the original trigger and by 22 more percent in comparison to the model retrained with the MESA-generated trigger.

| Backdoor | Before | | Retrain w/ rec-T | | Retrain w/ org-T | | NAD (Ours) | |
|---|---|---|---|---|---|---|---|---|
| Attack | ASR | ACC | ASR | ACC | ASR | ACC | ASR | ACC |
| BadNets | 100% | 85.65% | 4.96% | 81.23% | **3.91**% | **82.14**% | 4.77% | 81.17% |
| CL | 99.21% | 82.43% | 31.23% | 79.12% | 25.23% | 79.57% | **9.18**% | **80.34**% |

# E EXPERIMENTAL RESULTS OF HYPER-PARAMETER

We only give a rough estimate of $\beta$ for all the backdoor attacks in Figure 8 and it certainly provides better results by a more granular level of tuning.

---

[**]https://github.com/superrrpotato/Defending-Neural-Backdoors-via-Generative-Distribution-Modeling

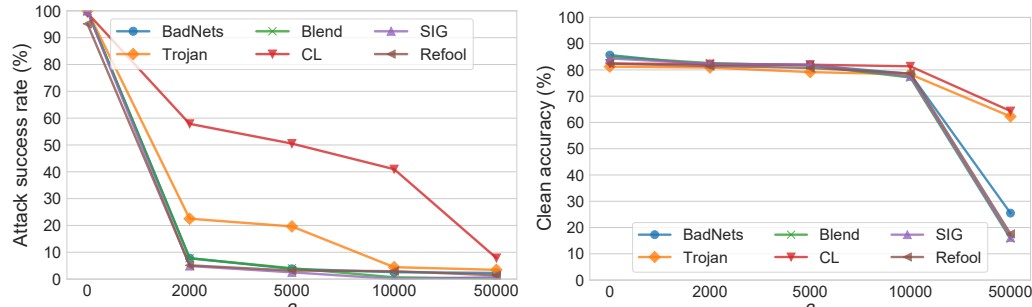

Figure 8: Parameter analysis: performance of our NAD approach under different $\beta$.

# F EXPERIMENTAL RESULTS OF DIFFERENT ATTENTION FUNCTIONS

We compare the performance of NAD under scenarios where 4 different attention functions, $\mathcal{A}_{mean}$, $\mathcal{A}_{mean}^2$, $\mathcal{A}_{sum}$, and $\mathcal{A}_{sum}^2$ are employed. We use the BadNets attack as our benchmark attack. Again, we evaluate the performance of NAD using two metrics, the ASR and the ACC, and the results are summarized in Table 7.

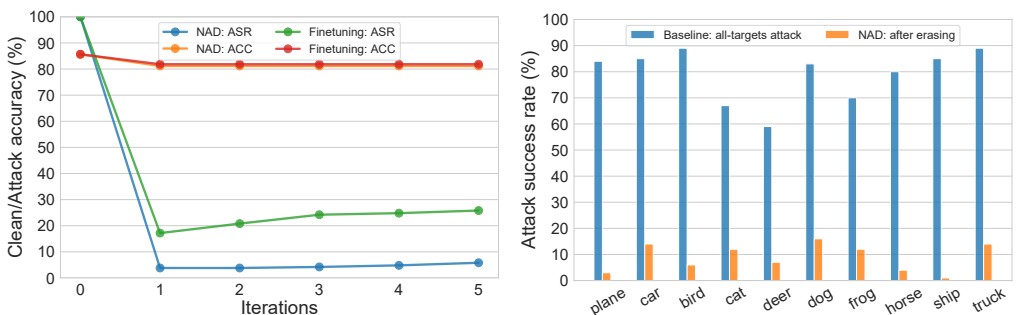

Figure 9: Iterative NAD and Finetuning against BadNets.

Figure 10: Erasing all-target BadNets attack.

Table 7: Performance of NAD using different attention functions against BadNets on CIFAR-10 with 5% clean data. ASR: attack success rate; ACC: clean accuracy. The best results are in **boldfaced**.

| Attention Function | $\mathcal{A}_{mean}$ | | $\mathcal{A}_{mean}^2$ | | $\mathcal{A}_{sum}$ | | $\mathcal{A}_{sum}^2$ | |
|---|---|---|---|---|---|---|---|---|
| | ASR | ACC | ASR | ACC | ASR | ACC | ASR | ACC |
| Baseline | 100% | 85.86% | 100% | 85.86% | 100% | 85.86% | 100% | 85.86% |
| Epoch 1 | 11.81% | 66.72% | 1.98% | 47.52% | 9.38% | 68.81% | 1.36% | 47.25% |
| Epoch 2 | 16.34% | 79.92% | 8.86% | 78.14% | 10.82% | 79.34% | 9.81% | 77.91% |
| Epoch 3 | 13.86% | 81.83% | 4.50% | 81.00% | 7.67% | 81.69% | 5.12% | 81.12% |
| Epoch 4 | 14.16% | 81.90% | 5.96% | 80.67% | 8.39% | 81.38% | 5.80% | 80.83% |
| Epoch 5 | 12.28% | 81.50% | 4.60% | 81.30% | 6.89% | 81.46% | **4.21%** | **81.55%** |

# G EXPERIMENTAL RESULTS OF ITERATIVE NAD

We evaluate whether NAD can be further improved with multiple iterations of distillation. In this experiment, we adopted the same configuration and set the iteration times to 5. Taking the BadNets attack as an example. The results in Figure 9 show that the attack rate has not been further reduced, and has even slightly increased by 2% in some cases. We hypothesize that the attentions of the backdoored neurons have been correctly aligned with the attentions of the benign neurons after a single-iteration of erasing. Whereas multiple iterations of distillation will make NAD refocus on the trigger pattern. Therefore, we believe that one-iteration of distillation is sufficient to guarantee the best result. Note that iterative Finetuning does not lead to further improvement over one-time finetuning neither.

## H   ERASING ALL-TARGET BACKDOOR ATTACKS

Unlike a single-target attack where the goal is to misclassify all backdoored images as one pre-specified target class, an all-target attack aims to misclassify every source class label as different ones (in our case, misclassify the original label $i$ as $(i + 1) \% 10$). In this experiment, we adopted the same settings (i.e. single target attacks on BadNets) to conduct all-target attacks on the WRN-16-1 network. We found that an all-target attack is a tougher task than a single-target attack. It is harder to attain a satisfactory ASR with an all-target attack. The results in Figure 10 show that NAD is able to reduce the ASR across all poison-classes (from 79% to 9.7%) effectively with only 5% of clean training data.

## I   FEATURE MAPS V.S. ATTENTION MAPS

A natural question to ask is: *why attention maps instead of feature maps?* This can be traced back to the field of knowledge distillation. Directly aligning the feature maps could lead to an information loss on the sample density in the space, and this could lead to a decrement in the distillation performance (Zagoruyko & Komodakis, 2017; Huang & Wang, 2017; Lopez et al., 2019). In the context of backdoor erasing, aligning the feature maps is not a good option because the backdoor neurons are only weakly, if not at all, activated by clean samples (Gu et al., 2019). In contrast, attention maps contain integrated information (see Equation 1) of both backdoored and benign neurons' feature maps, even when the neurons are not fired. (see Table 8). Figure 11 visualizes activation maps of a backdoored image on BadNets, Finetuned BadNets with 5% of clean training data, and BadNets erased by NAD with 5% of clean training data. The attention maps aggregated across the channels using 5 different attention functions are shown in Figure 12.

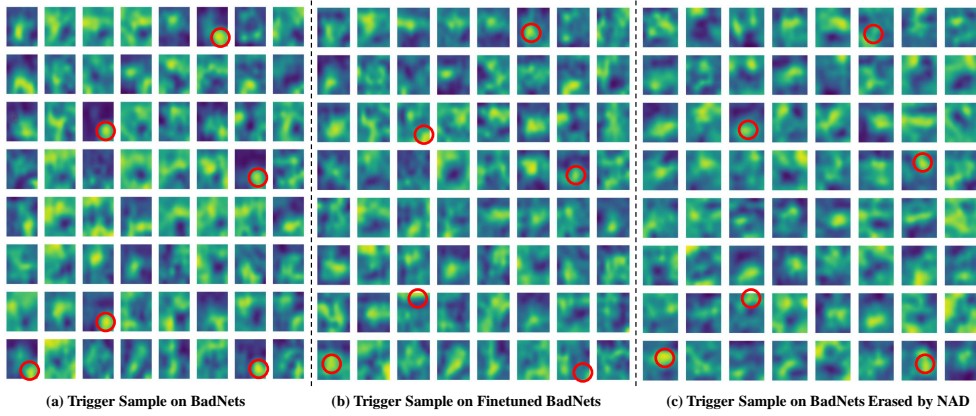

(a) Trigger Sample on BadNets     (b) Trigger Sample on Finetuned BadNets     (c) Trigger Sample on BadNets Erased by NAD

Figure 11: The activation map of one backdoored image at Group 3 of WRN-16-1 for (a) BadNets, (b) Finetuned BadNets with 5% of clean training data, and (c) BadNets erased by our NAD with 5% of clean training data. Each small patch is a channel (64 channels in total). The small red circles highlight the regions that are fired by the trigger pattern at different channels of the activation map.

Table 8: NAD using attention map versus activation map against BadNets.

|  |  | Before | Activation Map | Attention Map |
|---|---|---|---|---|
| CIFAR-10 | ASR | 100% | 98.44% | **3.81%** |
| (WRN-16-1) | ACC | 85.65% | 82.66 | 81.85% |

## J   OVERFITTING IN NAD

Here, we run NAD for a sufficiently long time (e.g. 20 epochs) to test if the student will eventually overfit to the finetuned teacher network. This experiment is conducted on CIFAR-10 against Bad-Nets and CL attacks. We also run the Finetuning defense as a comparison. Note that, the teacher network of NAD is only finetuned for 10 epochs, following the settings in Section 4.1. As shown

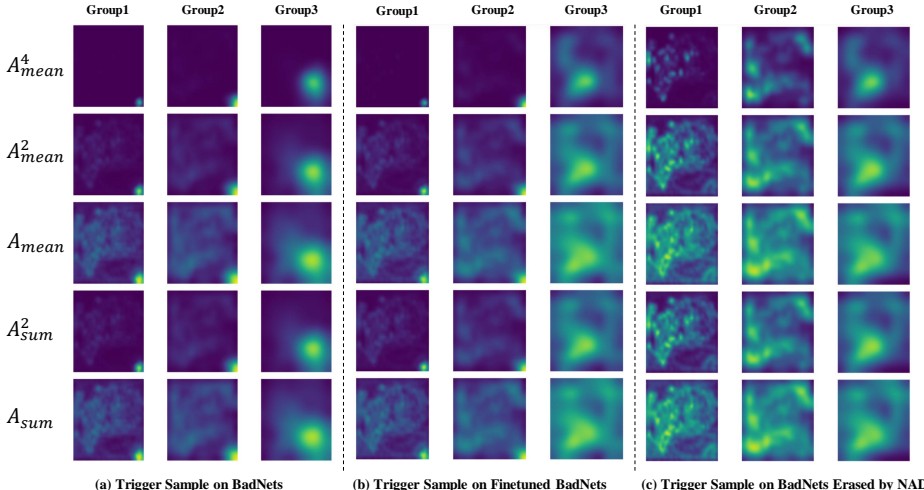

Figure 12: The attention maps derived by 5 different attention functions are shown for (a) BadNet, (b) Finetuned BadNet by 5% clean training data, and (c) BadNets erased by our NAD.

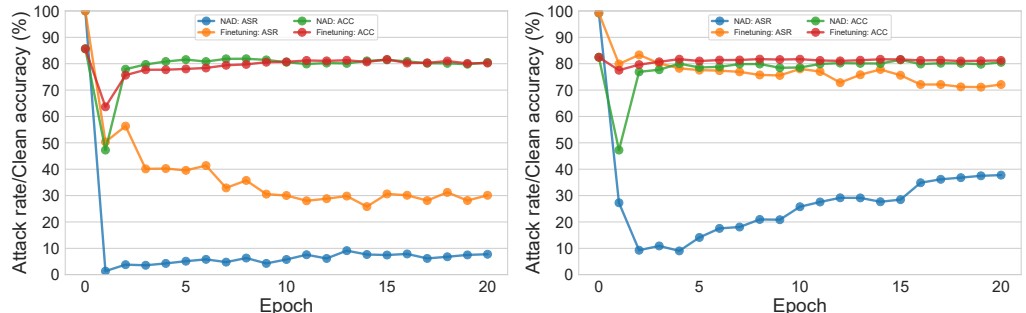

Figure 13: The learning curves (test ASR and ACC) of the NAD student network and a Finetuning network on CIFAR-10 against BadNets (left) and CL (right). In NAD, the student network tends to overfit to the teacher network, unless an early stopping is applied based on the validation ACC.

in Figure 13 (Appendix J), the student network of NAD can indeed overfit to the partially purified teacher network. However, this can be effectively addressed by a simple early stopping strategy: stop the finetuning when there are no significant improvements on the validation accuracy within a few epochs (e.g. at epoch 5). As shown by the green curves, the clean accuracy of NAD first drops, then quickly recovers and stabilizes at a high level within a few epochs. This also highlights the efficiency of our NAD defense as only a few epochs of finetuning is sufficient to erase the backdoor trigger.

## K  EFFECTIVENESS AGAINST ADAPTIVE ATTACKS

A backdoor adversary may attempt to construct a more stealthy backdoor trigger that does not cause obvious shift of the attention. To simulate this scenario, we design an adaptive version of BadNets on CIFAR-10 that attaches the trigger pattern at the center region of the image. Such an adaptive attack will only shift the attention close to the center region and has a weaker activation response. Since most of the CIFAR-10 objects are located at the center of the clean images, this adaptive attack may make the attention distillation much less effective. Figure 14 illustrates a few examples of backdoored images for this type of attack. We use a scaling parameter $\alpha \in [0, 1]$ to adjust the pixel value of a black-white square trigger pattern (all images are normalized into the range of $[0, 1]$). For example, for $\alpha = 0.2$, we scale pixel values of the trigger pattern $p$ to $p \times \alpha$. The results of our NAD against this adaptive attack are reported in Table 9. Our NAD method can still effectively erase the adaptive attack while maintaining high accuracy on clean data. Interestingly, Finetuning can only effectively erase the weaker trigger, and not as effective as NAD (especially in the case of $\alpha = 1.0$ attack). We conjecture this is because the center regions are more hard overwritten by the

clean images used for finetuning. We leave the exploration of more advanced adaptive attacks for future work.

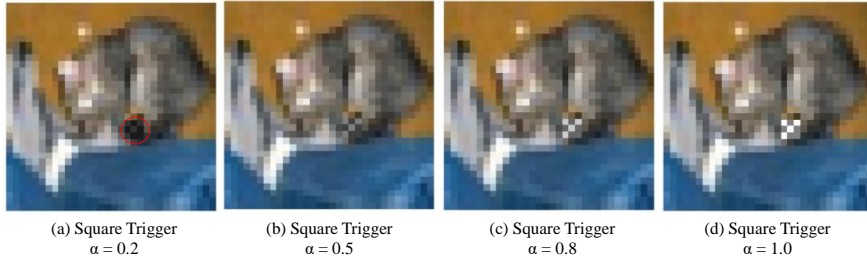

(a) Square Trigger
α = 0.2

(b) Square Trigger
α = 0.5

(c) Square Trigger
α = 0.8

(d) Square Trigger
α = 1.0

Figure 14: The triggers used by an adaptive BadNets attack against our NAD under different $\alpha$ (a scaling factor of the original black-white square). The trigger patterns are all placed at the center of the image.

Table 9: Performance of our NAD ($\beta = 2{,}0000$ for $\alpha = 1.0$ and $\beta = 1{,}0000$ for other $\alpha$) against an adaptive BadNets attack. ASR: attack success rate; ACC: clean accuracy. The best results are **boldfaced**.

| Square Trigger | Before | | Finetuning | | NAD (Ours) | |
|---|---|---|---|---|---|---|
| | ASR | ACC | ASR | ACC | ASR | ACC |
| $\alpha = 0.2$ | 99.85% | 82.11% | 7.51% | 79.26% | **4.92%** | **80.32%** |
| $\alpha = 0.5$ | 99.87% | 83.04% | 7.65% | 77.84% | **3.98%** | **78.91%** |
| $\alpha = 0.8$ | 99.97% | 82.85% | 12.65% | 79.91% | **4.08%** | **80.38%** |
| $\alpha = 1.0$ | 100% | 83.23% | 90.77% | 79.56% | **5.83%** | **80.41%** |

