# OpenReview forum: "Neural Attention Distillation: Erasing Backdoor Triggers from Deep Neural Networks"
_ICLR.cc/2021/Conference — ICLR 2021 Poster_

### Official Review · AnonReviewer4 · 2020-10-25
**Review #4**

**Rating:** 7
**Confidence:** 4

**Review:**

Summary:
This paper proposes a novel approach to erase backdoor triggers from neural networks through distillation. The defense method, called neural attention distillation (NAD), first finetunes the backdoored model on a set of clean data to get a teacher model. The second part of NAD then finetunes another copy of the original backdoored model (student) on the same clean data while minimizing the difference between the activation maps of the student model and teacher model. The teacher model’s weights are frozen in this distillation phase. The authors show that NAD outperforms or matches previous backdoor defenses over a range of attacks on the CIFAR-10 and GTSRB datasets. Further experiments are also conducted to show NAD’s performance under different conditions such as the percentage of clean data and configuration of teacher-student models.

Pros:
+Stronger empirical performance in defending against backdoor attacks versus previous methods.
+Solid experiments studying performance under different conditions such as clean data percentage and types of attacks.

Cons:
-The reason behind why NAD’s student model outperforms its teacher (Finetuning) model is unclear, leaving the intuition and basis behind NAD’s performance still not completely understood.
-NAD’s premise assumes the presence of clean and validated data, having limitations especially for training datasets with a large size.

Recommendation:
NAD’s better empirical performance over prior art and the comprehensiveness of experiments in this paper would be valuable in the effort to tackle the threat of backdoor poisoning. My key concern about this paper is the lack of discussion about why NAD’s student model could outperform its teacher (Finetuning) model even though both are finetuned on clean data and the student model is essentially mimicking the teacher’s activation maps and has the same model architecture. More discussion or theoretical analysis would make the intuition behind NAD more convincing. Moreover, the proposed method did not include a component to detect the presence of a backdoor poison or consider an adaptive attack scenario. Overall, I am still inclined towards accepting due to its good empirical results.


Comments & Questions:
Evaluating NAD on other model architectures might further show that NAD can generalize well.

Have you consider backdoor attacks targeting multiple classes? How would NAD perform?

What would happen to the required clean data size for the NAD defense to work if the dataset is of larger scale, such as imagenet? Would NAD still need 5% clean data, same, less or more?

--Update after rebuttal--
The reviewer thank the authors for the response. Most of the core issues have been addressed and scores have been updated accordingly.

---

> ### Author Response · Authors · 2020-11-19
> **Response to AnonReviewer4**
>
> Thank you very much for the valuable comments and suggestions. Please find our responses below for your questions.
>
> ---
> **Q1**: Evaluating NAD on other model architectures might further show that NAD can generalize well.
>
> **A1**: Thanks for the suggestion We have added the suggested experiments to Section 4.4 “Effectiveness of Different Teacher Architectures”. We tested 5 different architectures for the teacher network while fixing the architecture of the student network. The results show that our NAD framework generalizes well across different network architectures.
>
> ---
> **Q2**: Have you considered backdoor attacks targeting multiple classes? How would NAD perform?
>
> **A2**: Yes, we have considered the multi-class (all-target) case. The results are in Appendix H and Figure 10. Our method can effectively reduce the success rate of all-target BadNets on CIFAR-10 from 79% to 9.7%, with 5% clean finetuning data.
>
> ---
> **Q3**: How much clean data is required on a large-scale dataset like ImageNet.
>
> **A3**: Thanks for the question. However, to the best of our knowledge, it is still an open problem to develop effective backdoor attacks on large-scale datasets, and there is no existing backdoor attack on ImageNet. We plan to evaluate the performance of existing attacks when applied to ImageNet and test the effectiveness of our NAD method in our future work.

---

> > ### Comment · AnonReviewer4 · 2020-11-23
> > **Key concerns addressed by authors**
> >
> > The reviewer thank the authors for the response. Most of the core issues have been addressed and scores have been updated accordingly.

---

### Official Review · AnonReviewer1 · 2020-10-28
**Nice work. More clarification is appreciated.**

**Rating:** 7
**Confidence:** 5

**Review:**

This paper proposed neural attention distillation (NAD), a defense aiming to erase trigger effects from backdoored models with limited clean data. The idea is to leverage knowledge distillation by using a fine-tuned model as a teacher model and use NAD loss as a regularizer to train a student model by matching their attention maps.

Comparing to three existing defenses (finetuning, fine-pruning, and mode connectivity repairing (MCR)) over 6 different backdoor attacks, NAD shows significantly improved (lower) overall attack success rate while retaining similar accuracy on clean data. The authors also demonstrated that NAD training is more efficient than MCR. Some discussion on the teacher-student combination and the role of attention map is provided.

Overall, I find the defense results descent and the performance improvement significant. The authors also have done a thorough comparison across datasets, attacks, and defenses.

There are a few things I hope the authors can further clarify, and I believe the new results will add more value to this work. I am happy to increase my score if they are addressed.

1. Why is attention map regularization working, especially the fact that both the teacher and student models still carry backdoor effects (backdoored models w/ or w/o finetuning)? In the appendix the authors had some discussion on why using attention map over feature map for distillation loss. I am wondering does this imply feature maps are more vulnerable to trigger compared to attention maps? If so, since the tested attacks do not have the objective of deceiving the attention map (but will affect the feature maps), will the defense still be effective against an advanced backdoor attack that also aiming to deceive/bypass attention maps? In other words, knowing NAD will be used as a defense, it is possible that the attacker can devise an adaptive attack and design a backdoor model to make the distillation loss similar and mitigate the defense. I suggest the authors discuss and compare adaptive attacks to the proposed defense.

2. In the MCR experiment, it is not clear what two end models the authors chose to perform path connection. If the authros use backdoor finetuned model (teacher) and backdoor model (student) as two end models in MCR, will the defense results of MCR be different?

---

> ### Author Response · Authors · 2020-11-19
> **Response to AnonReviewer1**
>
> Thank you very much for the valuable comments. Please find our responses below for the concerns.
>
> ---
> **Q1**: Why is attention map regularization working? Are feature maps more vulnerable compared to attention maps? Evaluation on adaptive attacks.
>
> **A1**: There are two advantages of attention distillation over feature distillation: 1) **Integration**. The attention operators calculate the sum (or the mean) of the activation map over different channels (see Equation (1)). It can thus provide an integrated measure of the overall trigger effect. On the contrary, the trigger effect may be scattered into different channels if we use the raw activation values directly. Such a difference between the attention and the feature maps is visualized in Figure 11 and 12 in Appendix I. Therefore, aligning the attention maps between the teacher and the student networks is more effective in weakening the overall trigger effect than directly aligning the raw feature maps. 2) **Regularization**. Due to its integration effect, an attention map contains the activation information of both backdoor-fired neurons and the benign neurons. This is important as the backdoor neurons can receive extra gradient information from the attention map even when they are not activated by the clean data. Moreover, attention maps have lower dimensions than feature maps. This makes the attention map based regularization (alignment) more easily to be optimized than feature map based regularization. We have added a small subsection “Attention Distillation VS. Feature Distillation” in Section 4.3 to address this.
>
> While we agree that robustness analysis against an adaptive attack will make our results more complete, it is very challenging to develop such an adaptive attack. This is because backdoor attacks can only poison the training data and cannot intervent the training process. Therefore, the adversary has to trick the model to pay more attention to the trigger pattern, otherwise, the model may ignore the trigger pattern during training. It is still an open question whether a backdoor attack can succeed without causing an attention shift. Note that the attention shift is not necessarily towards a small patch, the Refool backdoor attack tested in our experiment has a very subtle attention shift (see Fig. 5 in the Refool paper). The evaluation on a simple adaptive attack with BadNets has been added to Appendix K.
>
> ---
> **Q2**: In the MCR experiment, it is not clear what two end models the authors chose to perform path connection.
>
> **A2**: As we stated in Appendix A “More Details on Defense Baselines”, we use the backdoored model as endpoint model t = 0 and t = 1. More results of using backdoored and finetuned backdoored models as endpoints are added to Table 4 in Appendix C.

---

> > ### Comment · AnonReviewer1 · 2020-11-23
> > **The authors' responses addressed my concerns**
> >
> > I thank the authors for the clarifications and additional experiments, which have addressed my concerns. I've increased my review score accordingly.

---

### Official Review · AnonReviewer3 · 2020-10-28
**Simple approach, great performance, and a comprehensive experimental setting**

**Rating:** 7
**Confidence:** 4

**Review:**

## Overview

The paper proposes a simple yet effective approach for purifying a neural network poisoned with backdoor attacks, AKA backdoor erasing. In short, the authors propose a two-step process: 1) fine-tuning the poisoned model on a small portion of clean data, which is a commonly used defense, and 2) treating the poisoned model as the student and the fine-tuned model as the teacher and performing attention distillation. The authors show the proposed approach's effectiveness by comparing to three commonly used techniques leveraging six state-of-the-art backdoor attacks on two datasets, namely GTSRB and CIFAR10.

## Contributions & Strengths

The paper is well-written, the approach is simple yet effective, the ideas are clearly communicated, and the article has a tremendous experimental section.

The contributions of the paper are as follows:

1. Introducing attention-distillation as a simple yet practical approach for erasing backdoors in a poisoned neural network.
2. Performing extensive studies to compare and validate the existing approach against state-of-the-art backdoor erasing methods
3. Providing substantive ablation studies that further clarify the contribution of each step of the proposed approach and its variations (for instance, in the iterative NAD).
4. The attacks used in the paper cover both patch-like ($3\times 3$ pixels) attacks as well as full image perturbation attacks.

Given the simplicity of the approach and its practical performance, I would expect the paper to be impactful and of interest to the community.

## Weaknesses

The main weakness of the paper, in my opinion, is the sensitivity of the approach to the regularization parameter $\beta$. Specifically, the optimal parameter choice relies on 1) knowing the type of attack and 2) having access to attacked images. It would be great if the authors can provide insights about choosing a good $\beta$ while being oblivious to the type of attack.

## Questions and comments for the authors

1. At first glance, the method is counter-intuitive. If I understand correctly, the teacher model is fine-tuned with the first term of the total loss in Eq (3), i.e., starting from the poisoned model, you fine-tune with cross-entropy loss on the clean data. Hence, the optimal solution for the first term of the loss in (3) is the teacher model.  The teacher model is also the minimizer for the second term in the loss. Therefore, I would expect the teacher model to be the optimal solution for minimizing (3). However, the paper shows that this is not the case, and optimizing (3) leads to a better model than the teacher model (i.e., the fine-tuned model). What am I missing?
2. Have the authors considered top-down attention mechanisms, like CAM, GradCAM, and GradCAM++, to calculate the attention maps in place of the norm-based attentions used in the paper? Could you provide any insights on this?
3. In "[Fooling Network Interpretation in Image Classification](https://openaccess.thecvf.com/content_ICCV_2019/html/Subramanya_Fooling_Network_Interpretation_in_Image_Classification_ICCV_2019_paper.html)" Subramanya et al. ICCV2019 propose adversarial patches that have a negligible effect on networks' attention (not in a backdoor setting). Given that the proposed method relies on attention, it would be interesting to see how it fairs against such attacks.
4. In all your ``clean accuracy' plots (e.g., Fig 2 right panel), could you please provide a zoomed-in version of the curves as well?

## Evaluation logic

Overall, I have a high opinion of this paper and appreciate the work the authors have put into writing a comprehensive article. I am scoring the paper as a 7. I would be happy to increase my score conditioned upon clarification of my questions and addressing the concerns.

## Post Rebuttal

I thank the authors for their response. I have two responses: 1) I still don't find the answer to my Q1 convincing; in particular, the 'filtering effect of distillation' mechanism requires more rigorous discussion, and 2) with regards to the ICCV2019 paper, I think the authors may have misinterpreted my point; my point here is that one could design backdoor attacks that do not affect the attention maps substantially and was wondering if the logic would hold for such attacks. However, I agree with the authors that the points go beyond the scope of the current paper and would be interesting for potential future work. In any case, I think the paper is a good contribution to the field and would still vote for accepting the paper.

---

> ### Author Response · Authors · 2020-11-19
> **Response to AnonReviewer3**
>
> Thanks for your thoughtful comments and interesting suggestions. Please find our responses below for your questions.
>
> ----
> **Q1**: The optimal parameter choice relies on 1) knowing the type of attack and 2) having access to attacked images. It would be great if the authors can provide insights about choosing a good $\beta$.
>
> **A1**: We agree that the optimal values of the $\beta$ parameter can be different across different attacks. A practical strategy is to increase $\beta$ until the clean accuracy (right subfigure in Figure 8) drops below an acceptable threshold. This can reliably find an optimal $\beta$, as increasing $\beta$ can consistently improve the robustness (left figure in Figure 8). We have added this discussion to Section 4.3 “Effect of Parameter $\beta$”.
>
> ---
> **Q2**: Eq. (3): Why the student network can be better than the teacher network?
>
> **A2**:  Thanks for your insightful interpretation. Indeed, the first term in Eq. (3) is to finetune the student network to keep the good classification performance, which will produce a model that has similar performance to the teacher network. The second term can further improve the attention of the student. Here, the key insight is that distillation has a filtering effect that can effectively remove the backdoor noise from the attention map, even if the target attention is not perfectly clean. In other words, the partially purified attention of the teacher network can lead to even better student attention. The filtering effect of distillation is enhanced by using the squared sum instead of the raw attention. We have an analysis in Appendix F and Table 6, where it shows $A_{sum}$ is better than $A_{mean} $ and $A_{sum}^{2}$ is better than $A_{sum}$. We have also added a small subsection “Why a finetuned teacher can purify a backdoored student?” in Section 4.4 to explain this insight.
>
> ----
> **Q3**:  Have the authors considered top-down attention mechanisms?
>
> **A3**: Thanks for the suggestion. The GradCAM and related attention is the attention of the network on the **input**, not intermediate layer attention. However, backdoor attacks implant triggers into the intermediate layers of the network. Since a backdoored network behaves normally on clean data, it is likely that the model will have normal attention on the clean images used for finetuning. This may limit the effectiveness of GradCAM attention based on distillation or regularization. We are happy to explore these mechanisms in our future work.
>
> ---
> **Q4**: Effect against "Fooling Network Interpretation in Image Classification" Subramanya et al. ICCV2019.
>
> **A4**: Thanks for the interesting suggestion. The patch attack proposed in the paper is a test-time adversarial attack, while our method is a training-time defense.  Although our method can effectively remove backdoor triggers from backdoored models, the purified models are still vulnerable to test-time adversarial attacks. We agree that it is interesting to explore strategies that can have both backdoor and adversarial robustness, for example, using adversarial finetuning. We will consider this as our future work.
>
> ---
> **Q5**: A zoomed-in version of the curves in Fig 2 right panel.
>
> **A5**: Thanks for the suggestion. We have added a zoom in version of the curves in Figure 2 right panel.

---

### Official Review · AnonReviewer2 · 2020-10-29
**Unconvincing results**

**Rating:** 6
**Confidence:** 4

**Review:**

This paper presents an empirical study on the backdoor erasing in CNN via teacher-student alignment of the attention maps.

S1: An unexpected way to erase backdoors without the need for additional information (provided that the experiments are done correctly).

W1: The experimental settings and results are highly doubtable.
W2: The findings lack theoretical support.

Detailed comments:

The training progress and the degree of overfitting of the teacher model are important but not clarified in the paper. In figure 4, it shows the four possible combinations of teachers and students. But I can’t understand why "4) C teacher and B student)" can work well too. Isn’t it prone to overfit? Maybe you can draw a learning curve for these 4 methods as well.

Please explain the possible source of information gain. For example: do the teacher and the student models use the same set of augmentation data? How does your approach compare with a baseline where only the teacher model is used and trained with both the data seen by the student and teacher models in your approach?

Please visualize the attention maps when the model is overfitting and when it is not. This helps people better understand the behavior of the NAD.

Please describe your data augmentation strategies in detail. What are their settings? Is the data augmentation methods used in each epoch or just at the beginning of the training process?

Overall, I am not convinced by the claims and results. The current results may be simply due to different degrees of overfitting or different data augmentation strategies. The author is encouraged to conduct more experiments to clarify this.

---

> ### Author Response · Authors · 2020-11-19
> **Response to AnonReviewer2**
>
> Thanks for your comments. Please find our responses below.
>
> ---
> **Q1**: The training progress and the degree of overfitting of the teacher model are important but not clarified in the paper. In Figure 4, why "4) C teacher and B student)" can work well?
>
> **A1**: We assume by “training progress”, the reviewer means the NAD finetuning process and the “overfitting” means the student network overfits the finetuned teacher network during NAD finetuning. We have added a paragraph at the end of Section 4.4 “Overfitting in NAD” to discuss this. We find that the student can indeed overfit to the partially purified teacher network. However, this can be effectively addressed by a simple early stopping strategy: stop the finetuning when there are no significant improvements on the validation accuracy within a few epochs (e.g. at epoch 5). Note that this also highlights the efficiency of our NAD defense as only a few epochs of finetuning is sufficient to erase the backdoor trigger.
>
> In Figure 4, the C teacher network is the model trained from scratch using 5% of clean training data (see Section 4.4). This means the attention of the network is clean. In other words, the network has learned to pay proper attention to the input image, although the model’s clean accuracy is low (right subfigure in Figure 4). In this case, the student can still overfit to the teacher network if it is overly finetuned.
>
> ---
> **Q2**: Do the teacher and the student models use the same set of augmentation data?
>
> **A2**: Yes. This information is provided in the last paragraph of Section 3 and in the “Defense Configuration” in Section 4. We have added more detailed descriptions of this setting at these two places.
>
> ---
> **Q3**: How does your approach compare with a baseline where only the teacher model is used and trained with both the data seen by the student and teacher models in your approach?
>
> **A3**: In our setting, the teacher and the student models use the **same** clean subset of data. So the suggested baseline is actually to finetune the backdoored model **two** or more times, which is also equivalent to finetuning for two times or more epochs. We find that iterative finetuning does not lead to further improvement over one-time finetuning. This result has been added to Figure 9 in Appendix G.
>
> ---
> **Q4**: Please visualize the attention maps when the model is overfitting and when it is not.
>
> **A4**: We assume by “the model is overfitting”, the reviewer means the student network overfits the teacher network. We have added the attention maps learned by the student network at each of the 10 finetuning epochs to Appendix I, along with a discussion on the overfitting effect in NAD at the end of Section 4.4. We also show the activation maps learned by the network when an activation (rather than attention) distillation is applied. As can be verified, using distilling activation can lead to severe overfitting to the teacher network, however, distilling attention does not.
>
> ---
> **Q5**: Please describe your data augmentation strategies in detail. What are their settings? Is the data augmentation methods used in each epoch or just at the beginning of the training process?
>
> **A5**: As we described in Section 4.1 “Defense Setting”, we use typical data augmentation techniques including random crop (padding = 4), horizontal flipping, and Cutout. For Cutout, we set the number of patches to be cut out of each image to 1 and each patch is a 3 × 3 square, which corresponds to the cutout parameters: n_holes=1 and length=9. This information is also provided in Appendix B. We have now added more descriptions of our settings in Section 4.1. The data augmentations are applied to each batch of training images at each training iteration, following a typical DNN training process. The same data augmentations are also applied to other finetuning-based baseline methods. Additionally, we have a comparison of our NAD method to just using data augmentations Cutout and Mixup in Appendix B.
>
> ---
> We are happy to provide more clarifications and run more experiments if there is anything still not clear or should you have other suggestions.

---

> > ### Comment · AnonReviewer2 · 2020-11-23
> > **My concerns have been partially addressed.**
> >
> > The reviewer thanks the authors for their response, which partially addressed my concerns. In particular, the authors have clarified the issue that the teacher model could overfit training data and how the corresponding student model evolves over epochs (Fig 13).  However, the possible reason behind the improvement made by NAD’s student model over the teacher model remains unclear. I have raised my score based on all the information provided.

---

> ### Author Response · Authors · 2020-11-23
> **Any additional questions?**
>
> Dear AnonReviewer2,
> Thanks again for reviewing our paper. Please let us know if you have any additional questions or require further clarifications. We are happy to address them before the rebuttal ends.

---

### Author Response · Authors · 2020-11-19
**Rebuttal Summary**

We sincerely thank all reviewers for their valuable comments and suggestions. We have made the following updates during the rebuttal.

---
+ Figure 2: added a zoom in view for the clean accuracy plot in Figure 2 (right panel).
+ Section 4.1: updated with more implementation details of data augmentations.
+ Section 4.3: added the new insights of attention distillation versus feature distillation.
+ Section 4.5: added the new empirical results for our NAD on different model architectures.
+ Section 4.5: added the explanations of using the finetuned teacher to purify the
student.
+ Section 4.5: added the discussion of overfitting in NAD.
+ Appendix C: added more results (Table 5) and analyses of Mode Connectivity Repair (MCR).
+ Appendix D: added the performance curves of iterative Finetuning in Figure 9 (left panel).
+ Appendix I: added the visualizations of activation maps versus attention maps for more understanding of our NAD (section 4.3).
+ Appendix J: added overfitting training plots for subsection “Overfitting in NAD” in Section 4.5.
+ Appendix K: added new empirical results to show the effectiveness of our NAD against a simple adaptive BadNets attack.

---
We have revised our paper according to all the valuable comments and please let us know if there is anything still not clear or any other suggestions.

---

### Decision · Program_Chairs · 2021-01-07
**Final Decision**

**Decision:**

Accept (Poster)

**Comment:**

This paper introduces neural attention-distillation; a new scheme for erasing backdoors in a poisoned neural network. The paper performs an empirical evaluation of their proposed method against  6 state-of-the-art backdoor attacks. The authors show that attention-distillation succeeds by using only a small fraction of clean training data without any performance degradation. In addition, the authors have provided ablation studies to clarify the contribution of each component in their proposed approach. Reviewers find the simplicity and effectiveness of the approach an important attribute that may lead this work to have a high impact in the field. The paper is well-written, and all reviewers rate it on the accept side. I concur with their opinions and comments and I recommend accept.